# FuseMax: Leveraging Extended Einsums to Optimize Attention Accelerator Design

Nandeeka Nayak*, Xinrui Wu†, Toluwanimi O. Odemuyiwa‡,
Michael Pellauer§, Joel S. Emer¶§, Christopher W. Fletcher*

*University of California, Berkeley. *nandeeka@berkeley.edu,cwfletcher@berkeley.edu*
†Tsinghua University. xr-wu20@mails.tsinghua.edu.cn ‡University of California, Davis. todemuyiwa@ucdavis.edu
§NVIDIA. mpellauer@nvidia.com ¶Massachusetts Institute of Technology. emer@csail.mit.edu

*Abstract*—**Attention for transformers is a critical workload that has recently received significant 'attention' as a target for custom acceleration. Yet, while prior work succeeds in reducing attention's memory-bandwidth requirements, it creates load imbalance between attention operators (resulting in severe compute under-utilization) and requires on-chip memory that scales with sequence length (which is expected to grow over time).**

**This paper ameliorates these issues, enabling attention with nearly 100% compute utilization, no off-chip memory traffic bottlenecks, and on-chip buffer size requirements that are independent of sequence length. The main conceptual contribution is to use a recently proposed abstraction—the *cascade of Einsums*—to describe, formalize and taxonomize the space of attention algorithms that appear in the literature. In particular, we show how Einsum cascades can be used to infer non-trivial lower bounds on the number of *passes* a kernel must take through its input data, which has implications for either required on-chip buffer capacity or memory traffic. We show how this notion can be used to meaningfully divide the space of attention algorithms into several categories and use these categories to inform our design process.**

**Based on the above characterization, we propose FuseMax—a novel mapping of attention onto a spatial array-style architecture. On attention, in an iso-area comparison, FuseMax achieves an average $6.7\times$ speedup over the prior state-of-the-art FLAT [16] while using 80% of the energy. Similarly, on the full end-to-end transformer inference, FuseMax achieves an average $5.3\times$ speedup over FLAT using 85% of the energy.**

## I. INTRODUCTION

Over the past few years, transformers [31] have emerged as the model architecture of choice for a wide range of machine learning applications, from natural language processing [11], [18], [29], [30] to computer vision [12], [20] to speech recognition [3], [14]. This rise has been accompanied by a corresponding wave of proposals for accelerating transformers in both software [7], [9], [10] and hardware [16], [34].

Fortunately, many of the layers (projections, fully connected layers, etc.) used by transformers look very similar to prior generations of machine learning models. Its resource-intensive tensor products can be described and evaluated with existing tensor algebra accelerator modeling tools [17], [22], [26], and many of the other layers (e.g., layer normalization) have negligible impact on performance and can be safely ignored.

However, attention [31]—usually described as a matrix multiplication, a softmax, and then another matrix multiplication—does not fit into either of these boxes. For

example, the softmax is both memory intensive (featuring low algorithmic reuse) *and* compute intensive (featuring exponentiation and division). Furthermore, attention's characteristics preclude many "free lunches" often used to improve efficiency for other DNN models. For example, because all tensors are a function of the model inputs, there is no opportunity to amortize memory access costs with an increased batch size. Additionally, since none of the operands can be computed before the inputs are given, compression/strength reduction techniques (e.g., quantization [13], [33], sparsity [21], [28], [32], etc.) must be applied dynamically, leading to more complicated algorithms and hardware designs.

To illustrate the difficulty in accelerating attention, consider the state-of-the-art accelerator for attention: FLAT [16]. FLAT uses fusion to reduce attention memory bandwidth bottlenecks on a spatial architecture (e.g., a TPU [15]). Specifically, FLAT maps attention's matrix multiplications to the 2D spatial array and softmax operations to a separate 1D array. While FLAT's design does make attention compute bound, it becomes compute bottlenecked in the 1D array (the softmax), causing severe under utilization of the 2D array. While one could add additional PEs to the 1D array, this results in commensurate area costs.

Making matters worse, FLAT requires that the entire vector over which the softmax is performed be buffered on chip. This vector is proportional to the sequence length, which is growing rapidly with time (e.g., Google reports 10 million length sequences in research, which would require 100s of MegaBytes to buffer [1]). When the vector/sequence length grows beyond allowable buffer capacity, FLAT is forced to spill, which contributes significantly to attention energy consumption and can even make attention memory-bandwidth bound.

**This paper.** We address the above challenges by proposing a novel spatial architecture – *FuseMax* – to accelerate attention, with particular emphasis on removing bottlenecks imposed by the softmax. Our architecture addresses all of the aforementioned issues associated with FLAT. FuseMax is compute bound, but provides almost 100% utilization of both the 2D and 1D arrays throughout the attention operation, without adding additional PEs to the 1D array. Additionally, FuseMax's on-chip memory requirements are invariant to sequence length and require no extra spills to memory regardless

of sequence length.

The technical core of the paper is two parts.

First, Section III uses the recently proposed *cascade of Einsums* abstraction [22] to describe, formalize and taxonomize the space of numerically stable attention proposals that appear in the literature. In a nutshell, an Einsum defines an iteration space over tensors and what computation is done on and between tensors at each point in the iteration space. A cascade of Einsums is a sequence of dependent Einsums that can be used to describe and specify a larger kernel.

While prior work [22], [25] provides a precise definition for Einsums, a major contribution in our work is to show how this definition can be leveraged to inform accelerator design. Specifically, we recognize that Einsums make explicit *precisely* what compute the cascade performs. We show how this can be used to inform trade-offs in designing the complicated, compute-intensive softmax kernel. Additionally, we recognize that the Einsum cascade makes explicit *precisely* what dependencies there are between Einsums. We show how this can be used to make non-trivial deductions about a kernel's allowed fusion granularity and algorithmic minimum per-tensor live footprint. The relationship between the live footprint and the buffer capacity, in turn, has implications for the required data movement. Given that an Einsum cascade is mapping/scheduling agnostic, *both* of the above observations provide insight given any possible scheduling of the cascade onto hardware.

In more detail, the latter provides insight into the number of times a given element of the an input must be revisited after visiting every other element of the input. Normally, one strives to choose a dataflow that exploits maximal reuse in a given element (or tile of elements) to avoid having to come back to it later. However, some algorithms preclude this strategy. For example, in a naïve implementation of attention, one must traverse the entire softmax input to build the softmax denominator *and only after that* can one revisit and scale each input (softmax numerator) by the denominator. Section III-C formalizes and generalizes this notion and shows that one can meaningfully divide the space of attention approaches based on the number of *passes* each cascade *must* perform, i.e., the number of times each element is (re)visited.

We note, our lower bounds on passes hold for all mapping choices, including application of fusion. For example, despite using fusion, FLAT employs a 3-pass cascade and its reliance on large on-chip buffering is a symptom of trying to avoid 3 passes-worth of DRAM traffic. Obviously, fewer passes is preferable; although, interestingly, we find that cascades with fewer passes can increase the required compute.

In the second part of the techical core (Section IV), we use the insights from Section III as a starting point to develop a novel mapping for attention that can be lowered to a spatial architecture. We call our architecture FuseMax. FuseMax adopts the attention cascade used in FlashAttention-2 [9]. However, despite using the cascade from FlashAttention-2, mapping this cascade to a spatial architecture instead of a GPU is non-trivial. We overcome the differences between the architectures and demonstrate a novel mapping for the cascade that achieves high utilization for entire transformer layers. Our architecture requires only minimal changes to a standard spatial architecture and is performance/energy robust to long sequence lengths (e.g., 1M tokens and beyond).

Finally, we evaluate FuseMax on BERT [11], TrXL [8], T5 [30], and XLM [18] and demonstrate a $6.7\times$ speedup on attention with $80\%$ of the energy and a $5.3\times$ speedup on the full end-to-end inference with $85\%$ of the energy relative to FLAT.

## II. BACKGROUND

In this section, we describe the concepts and terminology used in the remainder of the paper. This paper focuses on algebraic computations on tensors, where a *tensor* is a multidimensional array. A tensor's *rank* refers to a specific dimension of the tensor, while the tensor's *shape* is the set of valid coordinates for each of the tensor's ranks. We use the same symbol for the name and shape of the rank, i.e., the rank $M$ has shape $M$. A *fiber* is a set of points in a tensor with the same coordinate in all ranks except one.

An *Einsum* defines a computation on a set of tensor operands using an iteration space that specifies the set of points where the computations are performed [22], [25]. For example, we describe matrix-matrix multiplication (GEMM) computation with the following Einsum:

$$Z_{m,n} = A_{k,m} \times B_{k,n} \tag{1}$$

We follow the operational definition for Einsums given in [22]. In this work, we leverage the Extended General Einsums notation (EDGE) [25] (first developed for graph algorithms) to express the complex non-linear operations required for transformers.

TeAAL [22] introduces the concept of *cascades* of Einsums, which expresses directed acyclic graphs (DAG) of Einsum expressions as a sequence of sub-Einsums. For example, a matrix multiplication can be split into two Einsums, as follows:

$$T_{k,m,n} = A_{k,m} \times B_{k,n}$$
$$Z_{m,n} = T_{k,m,n}$$

An Einsum specifies the computation, while a *mapping* indicates how computation occurs in space and time on an accelerator [6], [26]. Mapping specifications include aspects such as loop order, partitioning, and work scheduling (sequential vs. parallel operations) [22]. Einsums in a cascade may be mapped independently or fused together to improve performance.

Transformer models generally follow the architecture defined in [31]. In this work, which addresses the impact of long sequence lengths during self-attention, we focus on the encoder architecture. Figure 1a gives an overview. As the sequence length grows, the relative importance of the different operations changes. Figure 1b shows that at shorter sequence lengths, the *weight-times-activation* "linear" layers are a larger fraction of the total required compute, while at long sequence lengths, the attention dominates. All additional required computation has negligible impact. In the next section, we focus

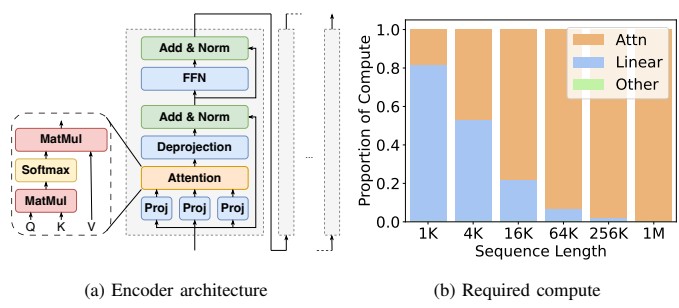

(a) Encoder architecture  (b) Required compute

Fig. 1: Overview of transformer encoder inference.

on describing attention more precisely, and use our analysis to understand prior work on efficient implementations.

## III. TAXONOMIZING ATTENTION AS EINSUM CASCADES

Our first contribution is to show how cascades of Einsums can be used to describe, taxonomize, and highlight trade-offs in the space of attention algorithms. The key insight is that cascades of Einsums provide a precise description of a kernel's compute requirements, the algorithmic minimum live footprint for each tensor, and the number of passes the algorithm *must* make through this live footprint.

### A. Redefining Attention's "Matrix Multiplications"

In the original transformer paper [31], the kernel was described with the following equation:

$$Attention(Q, K, V) = softmax(\frac{QK^T}{\sqrt{d_k}})V \qquad (2)$$

However, this equation says almost nothing about what the inputs $Q$, $K$, and $V$ look like or what iteration space needs to be traversed. We clarify these points by rewriting Equation 2 as a cascade of Einsums, with the exception of the softmax, whose cascade we will explore in Section III-B:

$$QK_{m,p} = \frac{1}{\sqrt{E}} \times Q_{e,p} \times K_{e,m} \qquad (3)$$

$$A_{m,p} = softmax(QK_{m,p}) \qquad (4)$$

$$AV_{f,p} = A_{m,p} \times V_{f,m} \qquad (5)$$

Here, Equations 3[1] and 5 look like matrix multiplications. Taking Equation 5 as an example, for each point in the iteration space $F \times M \times P$, we perform a multiplication using elements from two 2-tensors ($A_{m,p}$ and $V_{f,m}$) to produce a 2-tensor output ($AV_{f,p}$), which requires reducing across the inputs' shared rank $M$.

Equations 3-5 can be modified to refer to the full batched, multi-head self attention [31] by adding $B$ and $H$ ranks to all tensors. This changes the characteristics of the kernel. Adding the $B$ and $H$ ranks means that Equations 3 and 5 behave like many independent matrix multiplications instead

[1]In Equation 3, we also substitute $E$ for $d_k$ following the notation defined in Section II, where the shape of a rank is also its rank name.

of one monolithic matrix multiplication. The challenges with attention, described in Section I, follow clearly from this modification. Because *all* tensors contain a $B$ rank, the matrix multiplications are all unique to the specific batch's inputs. Therefore, none of these tensors can be computed before the inputs are given, and there is no data sharing between the different elements in the batch. To simplify notation, we assume the presence of the $B$ and $H$ ranks but omit writing them throughout the rest of paper.

### B. Softmax as a Cascade of Einsums

We now apply the same precise notation to the softmax. A softmax [4] over a 1-tensor is traditionally expressed with the following equation:

$$A_m = \frac{e^{I_m}}{\sum_k e^{I_k}} \qquad (6)$$

In the context of attention, this operation becomes two dimensional and can be expressed using the following cascade with input $QK_{m,p}$:

$$SN_{m,p} = e^{QK_{m,p}} \qquad (7)$$

$$SD_p = SN_{m,p} \qquad (8)$$

$$A_{m,p} = SN_{m,p}/SD_p \qquad (9)$$

For each point in the iteration space $(m, p)$, we exponentiate $QK_{m,p}$ to generate the softmax numerator ($SN_{m,p}$ in Equation 7), reduce $SN_{m,p}$ with addition to produce the softmax denominator ($SD_p$ in Equation 8), and finally, divide the numerator and denominator to produce the final result ($A_{m,p}$ in Equation 9).

*1) Improving Numerical Stability:* Because $e^{QK_{m,p}}$ can easily become extremely large, the above formulation suffers from overflow. Therefore, practical implementations [2], [27] often prefer the numerically stable variant that replaces Equation 7 with:

$$GM_p = QK_{m,p} :: \bigvee_m \max(\cup) \qquad (10)$$

$$SN_{m,p} = e^{QK_{m,p} - GM_p} \qquad (11)$$

and drop the $\frac{1}{\sqrt{E}}$ term when computing $QK_{m,p}$[2]. To compute the *global maximum*[3] $GM_p$, we reduce $QK_{m,p}$ with the operator max (instead of +)[4]. Notice that subtracting $GM_p$ from $QK_{m,p}$ in the exponent is equivalent to dividing by $e^{GM_p}$, and because the $\frac{1}{e^{GM_p}}$ term appears in both the numerator ($SN_{m,p}$ via Equation 11) and denominator ($SD_p$ via Equation 8), the result ($A_{m,p}$) stays the same. This construction improves numerical stability by bounding the values of the softmax numerator $SN_{m,p}$ (i.e., before the division) to the range $(0, 1]$.

[2]The $\frac{1}{\sqrt{E}}$ term was introduced to bound the magnitude of $SN_{m,p}$ [31]. Because the numerically stable softmax variant already accomplishes this, the scaling is often omitted [7], [9], [10].

[3]"Global" here refers to over the entire $M$ fiber.

[4]We use the reduction notation defined in [25]

## C. Optimizing Softmax Live Footprint and Memory Traffic

Einsums make explicit the dependencies between values produced and consumed across tensors in a cascade. We show how these dependencies can be used to make non-trivial deductions about a kernel's allowed fusion granularity and algorithmic minimum per-tensor live footprint. Because these deductions are made using only the cascade, they hold regardless of the choice of mapping or buffer capacity.

As a simple example, let us look at the softmax cascade Equations 7-9. If we want to minimize data traffic of $SN_{m,p}$, we need to choose a dataflow for each Einsum that keeps $SN_{m,p}$ stationary and fuses the three Einsums together. In other words, the dataflow must finish using the first element of $SN_{m,p}$ before moving onto the next. However, such a dataflow does not exist for this cascade. Any implementation must visit *every* element of a given $M$ fiber of $SN_{m,p}$ to compute $SD_p$ before it can revisit *any* element of that fiber to compute $A_{m,p}$.

We define a *pass* a cascade performs over a particular fiber of a particular rank and tensor to be a traversal of every element of that fiber. Each time an element *must* be revisited *after* visiting every other element of that fiber, there is an additional pass. For example, Equations 7-9 perform two passes over the $M$ rank of $SN_{m,p}$.

The number of passes a cascade performs is relevant because it restricts fusion schedules possible. While Einsums within a pass can be fused at will, producing and consuming a tile of the intermediate at a time, the two passes cannot be fused on the $M$ rank. Any implementation must visit all elements of an $M$ fiber of $SN$ to produce $SD$ before it can visit any of the elements of that fiber to produce $A$.

This analysis also provides a non-trivial lower bound on the tensors' live footprints. For example, the algorithmic minimum live footprint for tensor $SN$ is $M$. In other words, an architecture must either have enough buffer space to hold an entire $M$ fiber of $SN$ or spill and reload that fiber, incurring memory traffic proportional to the shape of $M$. We note that this analysis is mapping independent. There is no dataflow for this cascade that enables a smaller live footprint.

| 3-pass | 2-pass | 1-pass |
|---|---|---|
| PyTorch [27] | Tileflow [34] | FlashAttention [10] |
| TensorFlow [2] | Choi et al. [7] | FlashAttention-2 [9] |
| FLAT [16] | | |
| E.T. [5] | | |

TABLE I: Classifying prior attention algorithms.

We find that existing approaches to numerically stable attention can be classified as either 3-pass, 2-pass, or 1-pass cascades, where an $N$-pass cascade performs $N$ passes of a given $M$ fiber. See Table I.

## D. FuseMax Cascade Walkthrough

Next, we describe FuseMax's cascade (Cascade 1)—first proposed for FlashAttention-2. Note, despite the evidently increased compute relative to the 3-pass cascade, we will carefully design a mapping in Section IV to hide these overheads on a spatial architecture.

Initialization:

$$BQK_{m1,m0,p} = QK_{m1 \times M0+m0,p} \tag{12}$$
$$BV_{f,m1,m0} = V_{f,m1 \times M0+m0} \tag{13}$$
$$RM_{m1:m1=0,p} = -\infty \tag{14}$$
$$RD_{m1:m1=0,p} = 0 \tag{15}$$
$$RNV_{m1:m1=0,p} = 0 \tag{16}$$

Extended Einsums:

$$LM_{m1,p} = BQK_{m1,m0,p} :: \bigvee_{m0} \max(\cup) \tag{17}$$
$$RM_{m1+1,p} = max(RM_{m1,p}, LM_{m1,p}) \tag{18}$$
$$SLN_{m1,m0,p} = e^{BQK_{m1,m0,p}-RM_{m1+1,p}} \tag{19}$$
$$SLD_{m1,p} = SLN_{m1,m0,p} \tag{20}$$
$$SLNV_{f,m1,p} = SLN_{m1,m0,p} \times BV_{f,m1,m0} \tag{21}$$
$$PRM_{m1,p} = e^{RM_{m1,p}-RM_{m1+1,p}} \tag{22}$$
$$SPD_{m1,p} = RD_{m1,p} \times PRM_{m1,p} \tag{23}$$
$$RD_{m1+1,p} = SLD_{m1,p} + SPD_{m1,p} \tag{24}$$
$$SPNV_{f,m1,p} = RNV_{f,m1,p} \times PRM_{m1,p} \tag{25}$$
$$RNV_{f,m1+1,p} = SLNV_{f,m1,p} + SPNV_{f,m1,p} \tag{26}$$
$$AV_{f,p} = RNV_{f,M1,p}/RD_{M1,p} \tag{27}$$
$$\diamond : m1 \equiv M1 + 1 \tag{28}$$

Einsum Cascade 1: foo

We will start by expressing the partitioning of both of the inputs $QK_{m,p}$ and $V_{f,m}$ into M1 chunks of M0 elements each (Equations 12-13). This allows us to perform operations like maximum on individual $M0$ fibers, rather than on the whole tensor (Equation 17). The problem is, of course, that the local maximum is not necessarily the same for all $M0$ fibers and so will not just cancel nicely like the global maximum.

We resolve this by instead using the running maximum ($RM_{m1,p}$)—the global maximum of all inputs seen so far—instead of the local maximum. We recognize that $M1$ can also serve as an iterative rank, and iteratively build up $RM_{m1,p}$. After initializing $RM_{0,p}$ to $-\infty$ (Equation 14), we compute a new running maximum $RM_{m1+1,p}$ using the running maximum computed in the previous iteration $RM_{m1,p}$ and the new local maximum $LM_{m1,p}$ (Equation 18).

We can now use the running maximum to compute a local numerator $SLN_{m1,m0,p}$ (Equation 19), a local denominator $SLD_{m1,p}$ (Equation 20), and even the dot product result $SLNV_{f,m1,p}$ (Equation 21) using the partitioned $BV_{f,m1,m0}$ (Equation 13).

Now consider the softmax denominator. Eventually, we would like to reduce $SLD_{m1,p}$ into a 0-tensor, but because its values may have been computed with different maximums, we cannot simply use addition. Instead, by introducing a new running denominator $RD_{m1,p}$ with iterative rank $M1$, we can correct the old denominator $RD_{m1,p}$ to the new running

maximum $RM_{m1+1,p}$ and then perform the addition. Again, because $M1$ acts as an iterative rank for $RD_{m1,p}$, we must initialize the running denominator at the start of the computation to 0 (Equation 15). Then, at each point $m1$, the correction factor $PRM_{m1,p}$ allows us to correct the previous running denominator $RD_{m1,p}$ with the new maximum (Equation 23). In other words, $RD_{m1,p}$ is downscaled by $e^{RM_{m1,p}}$. $SPD_{m1,p}$ "switches" the downscaling factor on $RD_{m1,p}$ to $e^{RM_{m1+1,p}}$ by multiplying $RD_{m1,p}$ by $\frac{e^{RM_{m1,p}}}{e^{RM_{m1+1,p}}}$ ($PRM_{m1,p}$). Once $SLD_{m1,p}$ and $SPD_{m1,p}$ have the same maximum, they can be combined to produce the new running denominator $RD_{m1+1,p}$ (Equation 24). We can do the same to compute the running numerator-times-$V$ (Equations 16, 25-26).

Finally, $AV_{f,p}$ can be computed by dividing the final numerator-times-$V$ by the final denominator. By construction, at this point, $RNV_{f,M1,p}$ and $RD_{M1,p}$ are both downscaled by the same maximum $RM_{M1,p}$ (conveniently, also the global maximum) and can be correctly combined.

## IV. MAPPING ATTENTION ONTO A SPATIAL ARRAY

Based on the framework from Section III, we now describe FuseMax, an efficient mapping of a 1-pass attention cascade to a spatial array-style architecture. In this work, we use the same cascade as FlashAttention-2 [9].

The goal when mapping a cascade onto hardware is to fully utilize all available compute units. In our evaluation of prior work (Figure 4 and Section V), we observe that at short sequence lengths, the 2D PE array is under-utilized because it must wait for the 1D PE array to compute the softmax. At longer sequence lengths, both arrays are under-utilized since the workload becomes memory-bandwidth limited.

FuseMax's mapping addresses these issues to achieve full utilization on both the 1D and 2D PE arrays. We do so by (1) sharing the operations beyond multiply/accumulate (max/exp) between the 1D and 2D arrays and (2) ensuring that the workload is never memory-bandwidth limited by deeply fusing all Einsums in the cascade to restrict the live footprint to only what can be buffered on-chip. No matter the sequence length, our dataflow is never forced to spill any of its intermediates off-chip.

**Architecture.** We assume a standard spatial array-style architecture for our mapping. See Figure 2. We set parameters to match the cloud configuration in prior work [16]. Note, although both the 1D and 2D PE arrays perform exponentiation, we implement exponentiation with 6 sequential multiply-accumulate operations [24], [32] and therefore do not require a dedicated exponentiation unit.

**Fusion and Partitioning.** Prior attention accelerators [16], [34] explore fusing many of attention's loop nests together. However, because these accelerators all use multi-pass cascades, the algorithmic minimum live footprint of some tensors (e.g., $QK_{m,p}$) is $O(M)$, meaning that for long sequence lengths, intermediates cannot be buffered on chip.

FuseMax leverages fusion in conjunction with the 1-pass cascade to eliminate the memory traffic of these tensors, regardless of the sequence length. Specifically, we partition

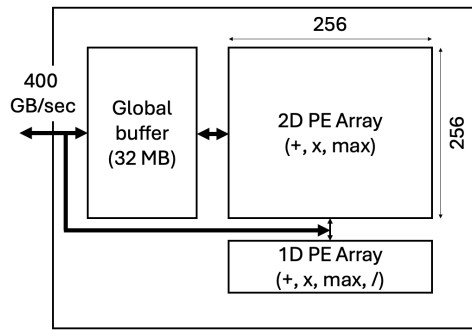

Fig. 2: Spatial array architecture assumed for FuseMax.

on both $M$ and $P$ (forming $M1, M0$ and $P2, P1, P0$), and maximally fuse all levels in the attention loopnest.

**Parallelization and Spatial Reduction.** While prior work implementing attention in hardware [16], [34] does utilize the 2D spatial array for the tensor products, it fails to do so for the softmax, choosing instead to use the 1D array. However, because there are far fewer total PEs in the 1D array than the 2D array, the softmax becomes a bottleneck. FuseMax improves utilization of the 2D spatial array by using it for both the tensor products and the exponentiation operator in the softmax. FuseMax parallelizes across the $M0$ and $P0$ ranks throughout the attention kernel. We set $M0 \times P0 = \#$ 2D Array PEs. The large spatial reductions required when parallelizing across the $M0$ rank are easily handled by the low-cost inter-PE communication network.

**Pipelining.** The dependencies between different Einsums in our cascade necessitate fine-grain pipeline parallelism to achieve high utilization of both the 1D and 2D spatial arrays. Figure 3 shows the waterfall diagram for FuseMax in the steady state. Time is broken into epochs. Each epoch performs the same set of tile-granular operations at specific tile-relative coordinates (given by $a, b, c, d$ in the figure). Across all epochs, the kernel evaluates all tiles and each Einsum is mapped to either the 2D or 1D array for all epochs (as shown in the figure).

A major design consideration when pipelining the mapping is how to overcome the latency of fills and drains to/from the spatial array. Without careful interleaving, the kernels suffers from low utilization due to long latency spatial reductions, whose results are required for future Einsums.

We address the above issues with two levels of interleaving. First, we interleave the construction of dependent tiles across epochs. This is reminiscent of software pipelining. For example, in Figure 3 the $d$-th tile of $BQK$ and $LM$ are completed in Epoch $i$ (as they correspond to a fill followed by a drain and can be easily pipelined). The $RM$ (which has to wait for the drain) for tile $d$ takes place *in a later epoch*. Instead, Epoch $i$ computes an earlier tile's running maximum $RM[c]$.

Second, we interleave the construction of certain tiles within an epoch at a fine (e.g., cycle-by-cycle) granularity. See the notation '$A|B$' in Figure 3. This is to ensure high utilization of both the 2D and 1D PE arrays at all times. In each cycle,

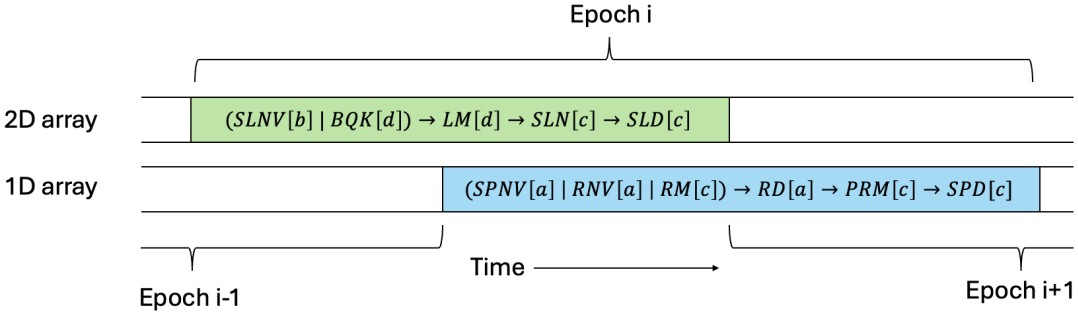

Fig. 3: FuseMax pipelining at a glance. Each tensor name (e.g., $SLNV$) corresponds to the Einsum used to compute that tensor (see Cascade 1). $a$, $b$, $c$ and $d$ denote tile-relative coordinates where $a < b < c < d$. If epoch $i$ produces tiles with coordinates $a, b, c, d$, epoch $i + 1$ produces tiles with identifiers $a + 1, b + 1, c + 1, d + 1$. And so on. '$A|B$' denotes 'computing tile $A$ is interleaved with computing tile $B$.' '$A \to B$' denotes 'computing tile $A$ is done before computing tile $B$.' The green and blue time periods making up an epoch take almost the same number of cycles.

a given PE in the 2D array computes a value for either $BQK$ or $SLNV$ and this alternates cycle by cycle. Each neighbor-neighbor link in the array is active in every cycle—carrying data for one of the two operation types. By interleaving $SLNV$ with $BQK$, the 1D PEs can concurrently compute $RNV$.

Putting everything together, as Section V will show, the above enables high utilization of all 2D and 1D array PEs.

## V. EVALUATION

In this section, we demonstrate how the FuseMax dataflow achieves improvements in both performance and energy relative to the state of the art, for both attention and the end-to-end transformer inference.

### A. Experimental Set-Up

First, we present the experimental set-up details common to all following subsections.

**Workloads.** We evaluate all accelerators and configurations using the same transformer models used by FLAT [16]: BERT-Base [11] (BERT), TrXL-wt103 [8] (TrXL), T5-small [30] (T5), and XLM [18]. We omit FlauBERT [19] because it uses the same hyperparameters as TrXL. We also note that though T5 is an encoder-decoder model, we only evaluate the encoder in this work. Following FLAT, we use a batch size $B = 64$ for all evaluations.

**Modeling with Timeloop.** We perform our evaluation using the tensor algebra accelerator modeling and design space exploration too Timeloop [26]. We use these tools to build models of the accelerator architectures and evaluate each Einsum individually. Results from individual Einsums are combined using heuristics presented in prior work for evaluating full cascades [22].

**Unfused Baseline.** We build the unfused baseline by combining the costs of three phases: $QK$ (Equation 3), the 3-pass softmax (Equations 10-11 and 8-9), and $AV$ (Equation 5). Because this baseline is unfused, each phase can be scheduled independently, but proceed sequentially and require outputs to be written to memory between phases. We use Timeloop to search for efficient mappings to perform $QK$ and $AV$.

Additionally, we model the softmax for the unfused baseline by allowing the accelerator to load the $M$ fibers of the input on-chip one-by-one (spilling if there is not enough space) before performing the compute. We model the memory traffic, compute, and energy required to perform all Einsums required for attention.

**FLAT Baseline.** Our main baseline is the state-of-the-art attention accelerator FLAT [16] (cloud architecture). Though we started with the FLAT authors' original code, we found and corrected a number of bugs with confirmation from the original authors. Beyond correcting the FLAT codebase, we created and validated a Timeloop model that reproduces the FLAT authors' (corrected) code to within $< 1\%$ error. However, the FLAT codebase does not model the cost to perform the softmax. Specifically, their model ignores the cost of data transfers (between any levels of the memory hierarchy) and uses $2^{30}$ 1D PEs. When comparing FuseMax and FLAT in this work, we augment our Timeloop model to model softmax correctly per the 3-pass cascade implicitly assumed by FLAT.

**Hardware parameters.** Figure 2 shows the selected hardware parameters, which were chosen to match FLAT's cloud accelerator. Also following FLAT, we use a 940 MHz frequency. We use Accelergy to confirm that, despite the changes to the PEs' functional units, the area difference between FLAT and FuseMax is $< 0.01\%$.

### B. Evaluating Attention

We now evaluate FuseMax to demonstrate the benefits it provides on the attention kernel by comparing it to the two baselines.

**Utilization.** Figure 4 shows the utilization of the 2D PE array. Because of the large amount of compute required for the softmax, both baselines achieve very poor utilization of this array. On the other hand, at long sequence lengths, FuseMax achieves almost 100% utilization. We observe that both baselines do achieve slightly higher utilization on XLM, which can be attributed to the higher intensity caused by a larger embedding dimension ($E/F$).

**Speedup.** Figure 5 shows that FuseMax achieves an average speedup of $10\times$ over the unfused baseline and $6.7\times$ over

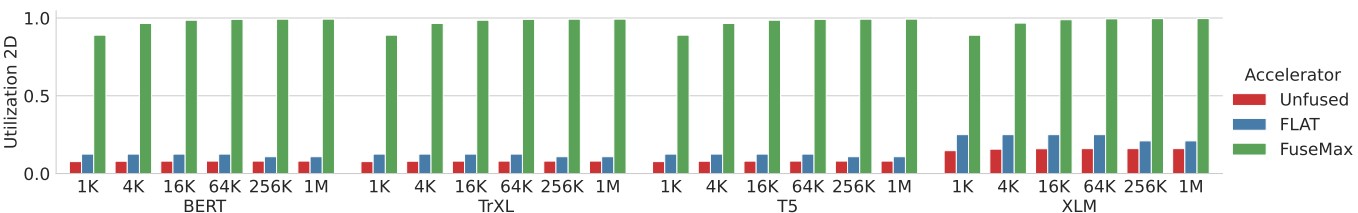

Fig. 4: Utilization of the 2D PE arrays on the unfused baseline, FLAT, and FuseMax.

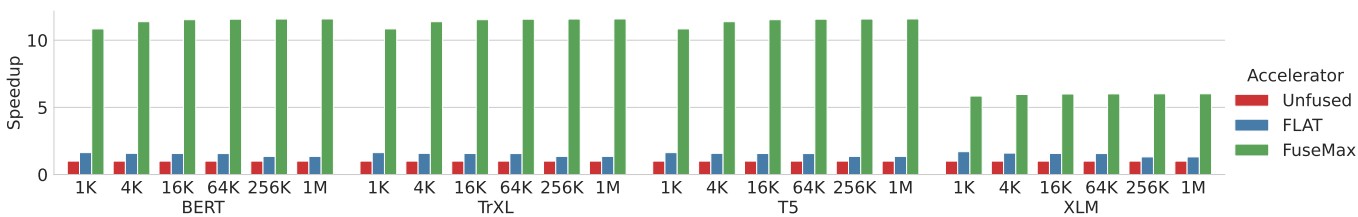

Fig. 5: Speedup of attention for FLAT and FuseMax over an unfused baseline.

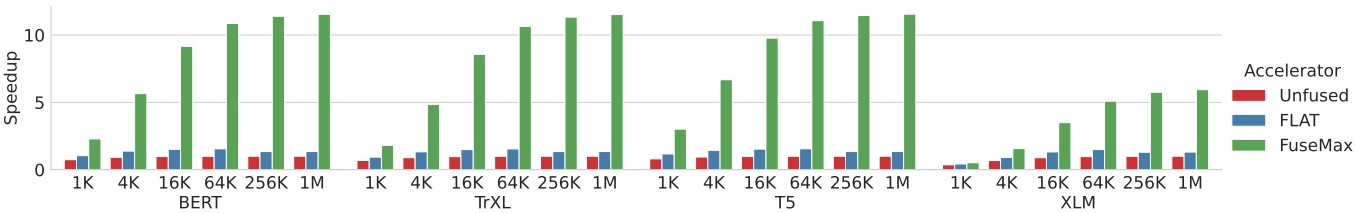

Fig. 6: Speedup of transformer inference on FLAT and FuseMax over an unfused baseline.

FLAT. We note FuseMax achieves lower speedup on XLM only because the baselines are able to achieve higher utilization of the 2D array on this transformer (Figure 4). An energy analysis can be found in the full paper [23].

## C. Evaluating Transformer Inference

To evaluate the benefits of FuseMax on end-to-end transformer inference, we include the other required linear layers (Section II). We use Timeloop to search for optimal mappings for these linear layers and use the same mappings for all three accelerator configurations. The attention modeling remains the same as Section V-B.

Figure 6 shows the performance improvement achieved by FuseMax. Across the sequence lengths tested, FuseMax achieves an average speedup of $7.6\times$ over the unfused baseline and $5.3\times$ over FLAT. As discussed in Section II, as sequence length grows, attention becomes a larger fraction of the total required compute. Therefore, at 1M tokens, FuseMax achieves an average $10\times$ speedup over the unfused baseline and $7.5\times$ speedup over FLAT. An energy analysis can be found in the full paper [23].

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
