# OpenReview forum: "FuseMax: Leveraging Extended Einsums to Optimize Attention Accelerator Design"
_iscaconf.org/ISCA/2024/Workshop/MLArchSys — MLArchSys 2024 OralPoster_

### Official Review · Reviewer_kuNX · 2024-05-24
**Paper about improving attention architecture in hardware**

**Confidence:** 3
**Rating:** 3

**Detailed Feedback And Questions For Authors:**

This paper presents important architectural contributions for avoiding memory bottleneck and improving the efficiency of attention hardware. They enable these contributions via revisiting math behind the softmax operation predominantly and via reconsidering the fusion behind einsum cascades generally. As a result, they were able to generalize softmax logic and achieve a high utilization of both 2D and 1D arrays of the attention hardware architecture.

Their primary point of comparison is FLAT paper from ASPLOS'23 which uses fusion of logit and attention operations to improve the computing efficiency. However, authors of the paper argue that FLAT's on-chip memory use grows unsustainably with tokens increasing which is prohibitive. Additionally, they also argue that with FLAT, 2D arrays remain still during softmax operations which, in turn, leads to a very low utilization rates. However, despite motivation and math explanations behind this FuseMax paper is clear, they spend very little time on actual hardware mapping and implementation of their solution, so it is very hard to evaluate the work which is the main reason for rejection for me.

Suggestions: less time for math, and more content for actual architectural implementation, more evaluation numbers on different aspects of the solution, more comments on limitation of the solution for a fair analysis, also some details of the FLAT paper can be explained a bit more for a clear vision of the competing solutions (I had to look up for the original paper to understand the FuseMax paper).

Nevertheless, thanks for your submission and important solutions in mind!

**Top Reasons To Accept The Paper:**

1) Speedup numbers look impressive.
2) Presented solutions like improving attention hardware and alleviating memory bottlenecks are important architectural contributions.

**Top Reasons To Reject The Paper:**

1) Despite the fact that paper presents impressive speedup numbers for important problems like alleviating memory bottleneck and increasing efficiency of attention hardware, its actual implementation is explained only at high level and very shortly at section IV. Hence, it is not clear how those high level details are implemented in the hardware.
2) Authors do not comment on potential drawbacks and limitations of the projects as well, which raises many questions on what are some pitfalls and tradeoffs.

---

### Official Review · Reviewer_TA4o · 2024-05-26
**Impressive results but lack of analysis**

**Confidence:** 4
**Rating:** 5

**Detailed Feedback And Questions For Authors:**

The paper provides impressive improvements over prior state of the art TPU-like spatial architecture. The paper uses the same cascade as proposed in FlashAttention2, and maps it onto a spatial architecture to improve compute utilization and achieve speedup.

However, the main contribution of this paper is not clear. Does the improvement in utilization (and the corresponding speedup) come purely from the fact that it uses the same 1-pass cascade as the one proposed in FlashAttention2, compared to the previous SOTA paper using a different 3-pass cascade?

I would also like much more details in Section 4 to support the claims made in the introduction. As an example, the introduction makes the following claim, "We show how this can be used to make non-trivial deductions about a kernel’s allowed fusion granularity and algorithmic minimum per-tensor live footprint." I did not find any analysis or explanation on this. Section 4 merely states "The workload is never memory bandwidth limited by deeply fusing all Einsums in the cascade to restrict the live footprint to only what can be buffered onchip." without clearly explaining how it is done (Does it come from FlashAttention2 or some new analysis?).

Another important factor on which I would like more details on is the claim made in the introduction: "However, despite using the cascade from FlashAttention-2, mapping this cascade to a spatial architecture instead of a GPU is non-trivial." What are the main differences and how are they solved? Is it solved just by sharing the non-linear operations between the 1D and 2D arrays? Are there any scheduling challenges involved in co-ordinating between these two arrays? What are the technical challenges involved in implementing the FlashAttention2 cascade on the spatial accelerator?

Additionally, most modern LLMs are decoder only architectures, so it would be interesting to explore the effect such casacdes on decoder based models. Will there be any differences in results and/or additional challenges involved in mapping these cascades for decoder based models?

The paper would benefit from a much deeper analysis of its insights and solutions. The final results are very impressive, and useful especially in the modern era where context lengths are huge. I understand this is a workshop paper, so there is a limitation on the number of pages, but I would like to see more text on analysis supporting the claims made in the introduction rather than explaining the prior work on how tensor operations can be expressed as einsum cascades.

**Top Reasons To Accept The Paper:**

- Impressive results over prior state of the art

**Top Reasons To Reject The Paper:**

- Lack of important research and technical details
- Ambigious claims

---

### Official Review · Reviewer_Xwzp · 2024-05-27
**Interesting approach to the use of einsums, but the rest of the paper lacks novelty and requires more investigation**

**Confidence:** 3
**Rating:** 4

**Detailed Feedback And Questions For Authors:**

**Summary:** FuseMax maps attention algorithms to a spatial array architecture by leveraging Einsum cascades to describe and optimize tensor operations.

**Strengths:**
* The paper introduces an interesting approach to use einsum cascades in accelerator design. In particular, cascade of einsums is used to describe, formalize and taxonomize the attention algorithms in the literature.
* The experimental results show substantial improvements over the state-of-the-art in both speed and energy efficiency.

**Weaknesses:**
* The paper lacks novelty. Most of the work builds upon the FLAT architecture.
* The paper assumes one configuration for the evaluation and provides limited information on the experimental setup, including the specific hardware used, software stack, and any optimizations applied during the evaluation.
* The writing needs to be improved. Especially, after Section III, the sections give the impression that they are cut. For example, Section V.A starts by "First, we present the experimental set-up details common to all following subsections." However, there is no other subsection in the remaining part of the paper.

**Questions for the authors:**
* Could you clarify how the foundational concepts from FLAT have been extended or modified to develop FuseMax?
* Could you provide more details on your implementation of FuseMax and FLAT on Timeloop and Accelergy, including the assumptions made?
* FLAT reports multiple configurations with various on-chip buffer sizes. Which one is used for the baseline evaluation?
* How would the utilization and speedup comparison change for various on-chip buffer sizes?

**Top Reasons To Accept The Paper:**

* The paper introduces an interesting approach to use einsum cascades in accelerator design. In particular, cascade of einsums is used to describe, formalize and taxonomize the attention algorithms in the literature.
* The experimental results show substantial improvements over the state-of-the-art in both speed and energy efficiency.

**Top Reasons To Reject The Paper:**

* The paper lacks novelty. Most of the work builds upon the FLAT architecture.
* The paper assumes one configuration for the evaluation and does not give enough details on the experimental setup.
* The writing needs to be improved.

---

### Official Review · Reviewer_YKea · 2024-05-27
**A systematic analysis of einsum to optimize attention acceleration**

**Confidence:** 5
**Rating:** 6

**Detailed Feedback And Questions For Authors:**

The paper proposes a systematic methodology to analyze and optimize operations in transformers by using Einsum cascades (an already proposed methodology). The use of Einsums to explicitly model data dependencies and derive minimum live footprints is a good contribution that can inform efficient hardware design and mapping. Based on the proposed methodology and analysis, the authors propose FuseMax architecture, showing improvements in performance and energy consumption over state-of-the-art methods, such as FLAT and FlashAttention. Here are few comments/questions:

(1) I understand that the authors had to condense the materials into four pages, which may have compromised clarity and detailed description of the method, but it would be great if the authors could add a more detailed explanation with illustrative examples, esp. for readers with less familiarity to these concepts (something that I hope the authors can fix during their potential presentation at the workshop).

(2) The models used for evaluation are limited and outdated. It would be interesting to see the effectiveness of FuseMax on a wide variety of models, including recent architectures.

(3) Same goes with the hardware evaluation, which was on a specific spatial array configuration. Performing sensitivity of FuseMax's performance to different hardware parameters would be valuable.

(4) Including more detailed breakdown of the performance gains, such as latency and energy consumption of individual components of the attention mechanism.

(5) Recent work shows sparsity/quantization can be productively used in attention mechanism. I am wondering if cascade of Einsum can be extended to cover such compression methods.

**Top Reasons To Accept The Paper:**

(1) The paper introduces a simple, yet new methodology to analyze and optimize attention algorithm using Einsum cascades (which is already introduced). The proposed definition provides clear insights into data dependencies and minimum live footprints.

(2) The authors propose a spatial architecture, FuseMax, and show promising speedups and energy efficiency compared to the state-of-the-art methods. FuseMax addresses key bottlenecks in attention computation (esp. for long sequences).

**Top Reasons To Reject The Paper:**

(1) The evaluation is limited to a few transformer models and a specific hardware configuration (I ack that limited number of pages was a factor here). Exploring a wider range of models and cofigurations improve the quality of the paper and strengthen the claims.

---

### Decision · Program_Chairs · 2024-05-30

**Decision:**

Accept (Oral/Poster)

**Comment:**

Congratulations! We are pleased to inform you that your paper has been accepted for presentation at MLArchSys 2024. We look forward to your participation at the workshop. Further details regarding the schedule and format will be provided soon. See you at the workshop!